# Unveiling the dynamics of antimicrobial utilization and resistance in a large hospital network over five years: Insights from health record data analysis

**Danesh Moradigaravand[1,2☯*], Abiola Senok[3,4☯*], Laila Al-Dabal[5], Hamda Hassan Khansaheb[6], Maya Habous[7], Hanan Alsuwaidi[3,6], Alawi Alsheikh-Ali[3,6]**

**1** Laboratory for Infectious Disease Epidemiology, KAUST Smart-Health Initiative and Biological and Environmental Science and Engineering (BESE) Division, King Abdullah University of Science and Technology (KAUST) Thuwal, Makkah 23955–6900, Saudi Arabia, **2** KAUST Computational Bioscience Research Center (CBRC), King Abdullah University of Science and Technology (KAUST), Thuwal, Makkah 23955–6900, Saudi Arabia, **3** College of Medicine, Mohammed Bin Rashid University of Medicine and Health Sciences, Dubai, United Arab Emirates, **4** School of Dentistry, Cardiff University, Cardiff, United Kingdom, **5** Infectious Diseases Unit, Rashid Hospital, Dubai, United Arab Emirates, **6** Dubai Academic Health Corporation, Dubai, United Arab Emirates, **7** Microbiology & Infection Control Unit, Pathology Department, Rashid Hospital, Dubai, United Arab Emirates

☯ These authors contributed equally to this work.
* abiola.senok@mbru.ac.ae (AS); danesh.moradigaravand@kaust.edu.sa (DM)













**Data Availability Statement:** We provide the code and aggregated data underlying the figures in the GitHub directory of the manuscript (https://github.

## Abstract

Antimicrobial Resistance (AMR) presents a pressing public health challenge globally which has been compounded by the COVID-19 pandemic. Elucidation of the impact of the pandemic on AMR evolution using population-level data that integrates clinical, laboratory and prescription data remains lacking. Data was extracted from the centralized electronic platform which captures the health records of 60,551 patients with a confirmed infection across the network of public healthcare facilities in Dubai, United Arab Emirates. For all inpatients and outpatients diagnosed with bacterial infection between 01/01/2017 and 31/05/2022, structured and unstructured Electronic Health Record data, microbiological laboratory data including antibiogram, molecular typing and COVID-19 testing information as well as antibiotic prescribing data were extracted curated and linked. Various analytical methods, including time-series analysis, natural language processing (NLP) and unsupervised clustering algorithms, were employed to investigate the trends of antimicrobial usage and resistance over time, assess the impact of prescription practices on resistance rates, and explore the effects of COVID-19 on antimicrobial usage and resistance. Our findings identified a significant impact of COVID-19 on antimicrobial prescription practices, with short-term and long-lasting over-prescription of these drugs. Resistance to antimicrobials increased the odds ratio of all mortality to an average of 2.18 (95% CI: 1.87–2.49) for the most commonly prescribed antimicrobials. Moreover, the effects of antimicrobial prescription practices on resistance were observed within one week of initiation. Significant trends in antimicrobial resistance, exhibiting fluctuations for various drugs and organisms, with an overall increasing trend in resistance levels, particularly post-COVID-19 were identified. This study

com/DaneshMoradigaravand/DubaiAMRProject). The raw data includes patients' sensitive information and can be obtained by any user with sufficient justification through communication with the Dubai Scientific Research Ethics Committee at DSREC@dha.gov.ae.

**Funding:** This work was supported by the King Abdullah University of Science and Technology baseline grant (BAS/1/1108-01-01 to DM). The funders had no role in study design, data collection and analysis, decision to publish, or preparation of the manuscript.

**Competing interests:** The authors have declared that no competing interests exist.

provides a population-level insight into the evolution of AMR in the context of COVID-19 pandemic. The findings emphasize the impact of COVID-19 on the AMR crisis, which remained evident even two years after the onset of the pandemic. This underscores the necessity for enhanced antimicrobial stewardship to address the evolution of AMR.

## Author summary

Antimicrobial Resistance (AMR) poses a global public health challenge exacerbated by the COVID-19 pandemic. Electronic health record data (EHR) presents an invaluable resource for dissecting public health trends and the effects of interventions in the context of antimicrobial resistance epidemiology and evolution; yet, this data has not been widely utilized. This study aimed to elucidate the impact of the pandemic on AMR evolution by leveraging population-level data from a centralized EHR of 60,551 patients with confirmed infections in a widespread hospital network in Dubai, UAE. We integrated structured and unstructured EHR data with microbiological laboratory data, COVID-19 testing information, and antibiotic prescribing data. Utilizing a broad range of analytical techniques such as time-series analysis, natural language processing (NLP), and unsupervised clustering algorithms, our research revealed substantial effects of COVID-19 on antimicrobial prescription rates and resultant resistance. The study also identified dynamic trends in antimicrobial resistance, demonstrating fluctuations for various drugs and organisms, with an overall increasing trend post-COVID-19. These findings underscore the critical need for enhanced antimicrobial stewardship strategies to mitigate the sustained impact of the COVID-19 pandemic on the evolution of AMR.

## Introduction

Antimicrobial resistance (AMR) is a growing global public health threat driven by widespread and inappropriate use of antimicrobials. The limited pipeline of novel antimicrobials highlights the need for increased emphasis on preventive measures to curb the spread of resistance. To this end, understanding population-level trends in antimicrobial utilization and resistance is essential to inform targeted preventive interventions. More recently, the evolution of AMR has been compounded in part by the COVID-19 pandemic. In the early phase of the COVID-19 pandemic, the limited knowledge about the newly emerged SARS-CoV-2 virus, inadequate access to rapid diagnostic tests, and clinical overlap of COVID-19 symptoms and bacterial pneumonia contributed to over-prescription of empiric antimicrobials for patients hospitalized with COVID-19 [1–5]. In addition, higher hospitalization rates among COVID-19 patients resulted in increased patient exposure to, and risk of acquisition of co-infections with hospital-associated pathogens that are often highly multidrug resistant [6–12]. On the other hand, the non-pharmaceutical interventions such as increased hygiene levels, decreased international mobilities, and diminished hospital procedures that were adopted during the pandemic may have mitigated against AMR in the short term [13,14].

Electronic health records (EHRs) are detailed patient records with demographic, lifestyle, symptom and signs, diagnosis, and outcome data stored in structured and free-text formats. Owing to the significance of this wealth of information, EHR data are increasingly being adopted for AMR research to enable surveillance of antibiotic prescription, consumption pattern and the associated diagnosis. EHR data have been used for assessing the outcomes of

infection, resistance trends, antimicrobial utilization as well as for interventional studies including point-of-care trials [15–17]. Analysis of EHR data has identified antibiotic use as a major risk factor for subsequent infections with antibiotic-resistant isolates [18,19].

Understanding AMR evolution within the context of COVID-19 requires population-level data integrating antimicrobial resistance levels with prescription data during this period. The difficulty of attaining such a dataset in most settings has hitherto limited the elucidation of such interplay. To understand the interplay between resistance trends and antimicrobial consumption and the effect of COVID-19 on such trends, we analyzed a large-scale centralized digitized EHR dataset, from the network of public healthcare facilities and diagnostic laboratories in Dubai, United Arab Emirates and dissected the dynamics of various population-level features of antimicrobial consumption and resistance. We examined the trends of antimicrobial consumption over time, known to show seasonal fluctuation, resulting from differences in the prevalence of infections and other clinical practices [20–22]. We analyzed the impact COVID-19 on the prescription differences and similarities for outpatients and inpatients. The groups may exhibit different prescription rates and patterns, due to the differences in disease severity [23,24]. We linked these patterns to the diagnosis notes and showed how COVID-19 has consistently affected prescriptions of a broad range of antimicrobials. Finally, we demonstrated the impact of antimicrobial prescriptions on resistance rates and quantified the association of antimicrobial resistance with mortality.

## Methods

### Ethics statement

This study was approved by Dubai Scientific Research Ethics (IRB number DSREC-05/2022_04).

### Database description

wData for 60,551 patients with confirmed bacterial infection was extracted from the centralized electronic platform for the Dubai Health Authority (DHA) which has now migrated to the newly created Dubai Academic Health Corporation. The dataset presented captures the patients seen in the public healthcare sector in Dubai during the study period. Based on publicly available data (2019–2021) reported in the Dubai Annual Health Statistics (https://www.dha.gov.ae/en/open-data), the total number of outpatients seen in Dubai across these years was 11,858,999 (2021), 8,730,496 (2020) and 11,653,351 (2019). Of these, 27.7% (2021), 33.2% (2020) and 20.5% (2019) were seen in the public healthcare sector. For inpatients, there were 402,666 in 2021 with 320,220 in 2020 and 332,393 in 2019. Of these, 22%, 27.9% and 21% were seen in the public healthcare sector in 2021, 2020 and 2019 respectively. This platform captures the health records of patients across the network of public hospitals (n = 4), primary care facilities (n = 13) and their associated diagnostic laboratories (n = 5). We analyzed data for all patients (inpatients and outpatients) diagnosed with any bacterial infection, based on laboratory confirmation of infection from samples sent to diagnostic laboratory as part of their clinical care for bacterial identification and drug susceptibility testing, between 01/01/2017 to 31/05/2022. We curated and linked the following datasets using patients' unique medical record number.

### Antibiogram

The antibiogram data of isolated organisms from each patient against a panel of antimicrobials routinely tested in the DHA diagnostic laboratories. The diagnostic laboratory follow the Clinical and Laboratory Standards Institute (CLSI) M100 guideline (https://clsi.org/standards/

products/microbiology/documents/m100/) which is updated annually for the interpretive categories of disk diffusion or minimum inhibitory concentrations (MIC) breakpoints derived from the Vitek 2 automated system or E-Test (bioMérieux, Marcy l'Etoile, France). In addition, the annually updated EUCAST guideline is applied for the interpretation of the Colistin MIC determination which is performed by broth microdilution method. The data included 1,980,519 tests for 45440 patients. We further classified the organisms and antimicrobials into broader groups and classes to obtain insights into shared resistance mechanisms. For analysis, we classified test results into resistant and susceptible groups. The laboratory dataset also included patients' demographic data, including age, gender, and ethnicity.

## Microbial genotyping

This comprised of molecular typing data for the presence or absence of major resistance genes, including extended spectrum beta lactamase (ESBL), carbapenem resistance, methicillin resistance, and vancomycin resistance genes. Molecular typing data was available for 35% of the patients diagnosed with a bacterial infection. In total, 114,254 test results for 2008 patients were analyzed. The list of organisms is provided in the Supplemental data on GitHub.

## COVID-19 testing

The dataset encompasses the results of comprehensive COVID-19 RT-PCR testing, consisting of 144,636 tests conducted on 35,050 patients. The COVID-19 testing was carried out on symptomatic and asymptomatic patients. The dataset includes information about the test dates and patient demographics.

## Electronic Health Record (EHR)

We retrieved the EHR clinical and prescription data for all inpatients and outpatients with a positive bacterial infection. In total, we analyzed the records for 60,551 patients. The data is composed of both structured and non-structured records. The structured data included patients' encrypted ID, registration dates, demographic and clinical data including age, gender, body mass index (BMI), date of hospitalization, clinical outcome and mortality. The nonstructured text data included diagnosis notes reporting comorbidities for each patient clinical encounter. The EHR data contained prescription records of antimicrobials for individual inpatients during hospitalization and for individual outpatients at each visit. The inpatients (outpatient) prescription data included prescriptions for 29,238 (47,226) patients in 50,166 (251,909) encounters. The text corpus for prescription notes consisted of 50,090 and 249,865 documents for inpatients and outpatients, respectively.

## Time series data analysis for prescription

To facilitate the analysis, we aggregated the daily prescription data into weekly time intervals. We computed the relative prescription rate for each antimicrobial as the number of prescriptions of the antimicrobial divided by the total number of prescriptions during each week for the patients with a confirmed infection. This metric shows the relative importance of an antimicrobial in comparison with other antimicrobial in each week. The aggregation of the metrics from different weeks yielded time data for drugs. We converted the data points into time series with yearly trends functions in the stats package in R and then decomposed a time series into seasonal, trend and irregular components using moving averages. We then extracted the trends of the time series (i.e., a pattern in data showing how a series has changed to relatively higher or lower values over tim).

## Integration with COVID data and time intervention analysis on antimicrobial consumption

We estimated the association between antimicrobial consumption and COVID-19 infection by integrating data from the consumption of antimicrobials for inpatients and outpatients with the record of SARS-CoV-2 infection. We considered the patient as COVID-19 positive if they had at least one positive RT-PCR test result. For the drugs for which a significant link between consumption and COVID-19 was found, we conducted a causal intervention analysis to approximate the causal impact of COVID-19 on the prescription rate. The analysis allowed estimating the impact of the intervention (COVID-19 pandemic in our study) on the time-series data for the type of time-series data that cannot be reproduced without the intervention. The estimated casual effect is compared with the scenario of what would have happened had the intervention not occurred. We employed the **Causal Impact** library in R [25], which allows for analysis of intervention effects by using a separate series (one which is not affected by the intervention) as a covariate. The package uses a structural Bayesian time-series model to estimate the *pointwise* causal effect. This shows how the response metric might have evolved after the intervention if the intervention had not occurred. The effect is estimated for pre- and post-periods. We defined these periods assuming 01/01/2020 as the onset of the pandemic and estimated the causal impact for different period lengths (weeks) before and after COVID-19 onset. The effect is measured with respect to a counterfactual time series, in which no effect is assumed. To compute this baseline time series, we first obtained the standard deviation and mean of the time series between COVID-19 onset and the period before the COVID-19 onset. We then used the ARIMA function to generate a time series with no seasonality or trend. We reported the relative effect of intervention value from the model, corresponding to the average weekly difference between actual values and predicted values in the post-COVID-19 period, transformed in percentage.

## Diagnostic term text analysis

The non-structured EHR notes for each patient with a bacterial infection were analysed to assess the importance of terms for antimicrobial prescriptions. We first created a separate text corpus for each antimicrobial, containing all diagnostic notes associated with the prescription of the antimicrobial. A text processing pipeline was then developed using the tidytext package in R to cleanse the text by removing special characters and stop words. The BC5CDR (Bio-Creative V CDR corpus) disease name entity recognition which is a library of 1500 PubMed articles with 4,409 annotated chemicals, 5818 diseases and 3116 chemical-disease interactions was used [26]. The term-weighting schemes of frequency–inverse document frequency (tf-idf) was computed, showing the importance of a word to a particular corpus in a collection of corpora. The tf–idf value rises in proportion to the number of times a word appears in the document and is offset by the number of documents in the corpus that contain the term, which helps to account for the fact that some words appear more frequently than others.

## Antibiogram analysis and resistance level

We analyzed the antibiograms from diagnostics laboratories for all patients with a confirmed infection throughout Dubai. The antibiogram data included patients' demographic data and the drug susceptibility testing results against different groups of antimicrobials. We assumed two classes of resistance phenotypes: resistant and broad susceptible, including susceptible and intermediate phenotypes. We aggregated daily rates into the weekly rate of resistance by dividing the number of resistance test results by the total number of resistant and susceptible tests

for each antimicrobial and the organism. This allowed to account for the variation in the total number of tests. We then extracted the trend using the same above-mentioned method.

## Correlation between antimicrobial consumption and resistance

We employed the cross-correlation function (CCF) for modeling the cross-correlation of two univariate time-series of the antimicrobial weekly consumption rate and resistance level. The antimicrobial weekly consumption rate for each specific antimicrobial was calculated by dividing the count of prescriptions for that antimicrobial by the total number of patients in the week. The consumption rate corresponds to the antimicrobial consumption per patient. The purpose of the analysis is to find linear dynamic relationships in time-series data, generated from stationary processes. The sample cross-correlation function (CCF) identifies lags or leads of the two time-series, for consumption, denoted by $C_t$ and for resistance level, denoted by $R_t$. We examined if the series $R_t$ is related to past lags of $C_t$, using the CCF. We defined the sample CCF as the set of sample correlations between $R_{t+w}$, where $w$ designates weeks, and $C_t$, for $w = 0, \pm1, \pm2, \pm3$, and so forth. In situations where one or more $R_{t+w}$ values, with negative (positive) $w$, were found as predictors for $C_t$, we classified this as leading (lagging) relationships from $R$ to $C$. Using the ccf function as part of the acf package in R, we computed the cross-correlation between the time series for different lag weeks for the same drug or major drugs belonging to the same class, e.g., azithromycin and erythromycin, wherever the data for consumption and resistance for the same antimicrobial was available.

## Time series classification for antimicrobial consumption and resistance

We classified the trends extracted for the consumption of antimicrobials for inpatient and outpatient groups, as well as the resistance levels, to identify major trends in consumption and resistance. We examined this for the top ten most frequent antimicrobials tested for the top most frequent organisms. We employed the mclust R package for model-based clustering and classification based on a finite normal mixture modelling [27]. The classification function provides functions for parameter estimation via the EM algorithm for normal mixture models with a variety of covariance structures. We tried eight different types of models with cluster numbers between two and nine. We selected the best clustering based on the highest value for the Bayesian Information Criterion (BIC). We then used principal component analysis (PCA) for the visualization of the clustering results.

## Resistance effect on mortality

The mortality outcome of the patients was integrated with the resistance rate to approximate the potential mortality risk associated with infection by resistant strains. The analysis was conducted for each organism and the most prescribed antimicrobials for each group of patients. We designated the alive group as the reference and resistance as exposure in the analysis. The odds ratio was calculated by median-unbiased estimation (mid-p) and confidence intervals (95% confidence interval) by mid-p methods, using the odds-ratio function from the epitools package in R [28]. The results from the odds-ratio calculation were confirmed with binary logistic regression analysis. We considered the mortality outcome as a binary response variable and assessed the significance of the effect of resistance on reported death for antimicrobial $j$ and for infection of patients with organism $i$ in combination with the confounding effects of age, gender, ethnicity, BMI, hospitalization length, hospitalization frequency, and diagnostic

notes as follows:

$$Death\ {}^{organism_i}_{antimicrobial_j} \sim Resistance^{organism_i}_{antimicrobial_j} + age + ethinicty + gender + BMI + Hospitalisation\ Length + Hospitaloisation\ frequency + \sum_{j=1}^{121} term_j$$

In above *Death* is the binary feature corresponding to the mortality status of death or alive. Ethnicity and gender were factorized and converted into numerical values. Hospitalization length represents the total number of days that the patient spends in the hospital, while hospitalization frequency is the number of times the patient was admitted to the hospital. Resistance is a binary feature showing the resistance status of the colonizing organism, i.e. resistant or susceptible. We included the top 0.1% of the diagnostic terms (*terms*) (121/147455) features in the equation. These features included the binary encoded one and zero corresponding to the presence or absence of features related to the infection type, infection site, and comorbidities, reported in the notes. The regression coefficient associated with the resistance for each organism *i* and antimicrobial *j* was extracted. This coefficient corresponded to the change in log odds of mortality of having the outcome when switching from susceptible to resistant states.

We also conducted survival competing risk regression (the Fine-Gray regression) [29,30]. In this analysis, death was considered the event of interest, and the same predictors that were employed in the logistic regression model were evaluated as potential factors influencing the time until death. The survival equation took the form:

$$Surv(age, Death\ {}^{organism_i}_{antimicrobial_j}) \sim Resistance^{organism_i}_{antimicrobial_j} + ethinicty + gender + BMI + Hospitalisation\ Length + Hospitaloisation\ frequency + \sum_{j=1}^{121} term_j$$

In above equation, Surv denotes the survival curve, in which death is the event and age as the time to the event. We employed the 'crr' function for Fine-Gray regression in the ' tidycmprsk' R package and reported the coefficient for the Resistance feature (competing risk estimate). The above processes were then repeated for the presence of resistance genes and mortality, by replacing the presence and absence of the resistance gene from genotyping data with the above resistance term.

## Results

### Observed pattern among In-patients and out-patients

The analysis of prescription and diagnostic laboratory data revealed that antimicrobials were prescribed at an average rate of 1.91 (with a minimum of 1.82 in 2022 and a maximum of 2.04 in 2020) and 1.30 (with a minimum of 1.28 in 2018 and a maximum of 1.32 in 2020) prescriptions per encounter for the inpatient and outpatient groups, respectively. A comparison of the most commonly prescribed antimicrobials showed a high degree of overlap between the inpatient and outpatient groups, with 67% (74 out of 110) of the antimicrobials being shared (S1A Fig). This finding is in line with a previous report of a comparable large-scale multi-center study in China, wherein the similarity in prescriptions between inpatient and outpatient groups was attributed to shared clinical practices and infection management [31]. The most frequently prescribed antimicrobials for both groups included antibiotics for the respiratory tract, pelvic, and urinary tract infections. The most frequently prescribed antimicrobials among outpatients with an infection were amoxicillin/clavulanate, cefuroxime, clotrimazole,

fusidic acid, and azithromycin (S1A Fig). Meanwhile, the highest number of prescriptions among inpatients were for metronidazole, amoxicillin-clavulanate, cefuroxime, and ciprofloxacin (S1B Fig). The ranking of the most commonly prescribed antimicrobials was mostly similar across both inpatient and outpatient prescriptions (S1B Fig). This pattern aligns with the prescription practices in other high-income countries e.g. in USA [32], where amoxicillin, ciprofloxacin, and azithromycin are among the most frequently prescribed drugs. These results suggest that the findings from our dataset may be applicable to other settings as well.

## Seasonal pattern of antimicrobial prescription

We observed seasonal patterns in antimicrobial prescription counts during the pre-COVID-19 period (i.e. before 2020) for the top 16/20 (inpatients) and 17/20 (outpatients) prescribed antimicrobials, with peak prescriptions occurring between September and March for both inpatient and outpatient groups (Fig 1). The average percentage increase in values between the peak month and the minimum month for these antimicrobials was 42% and 52% during the September to March period, resulting in a peak-to-trough ratio (PTTR) of 1.42 and 1.56 across the most commonly prescribed antimicrobials for inpatients and outpatients, respectively (Fig 1). Ethambutol and Pyrazinamide were exceptions, with peak occurrences happening inversely during the summer before the pandemic (Fig 1). Text analysis of admission diagnostics revealed that these drugs were primarily used to treat tuberculosis, for which a spring-summer prevalence has been reported [21]. After the onset of COVID-19, the seasonal distribution of prescriptions underwent a significant change, with the peak for the top 14 and 13

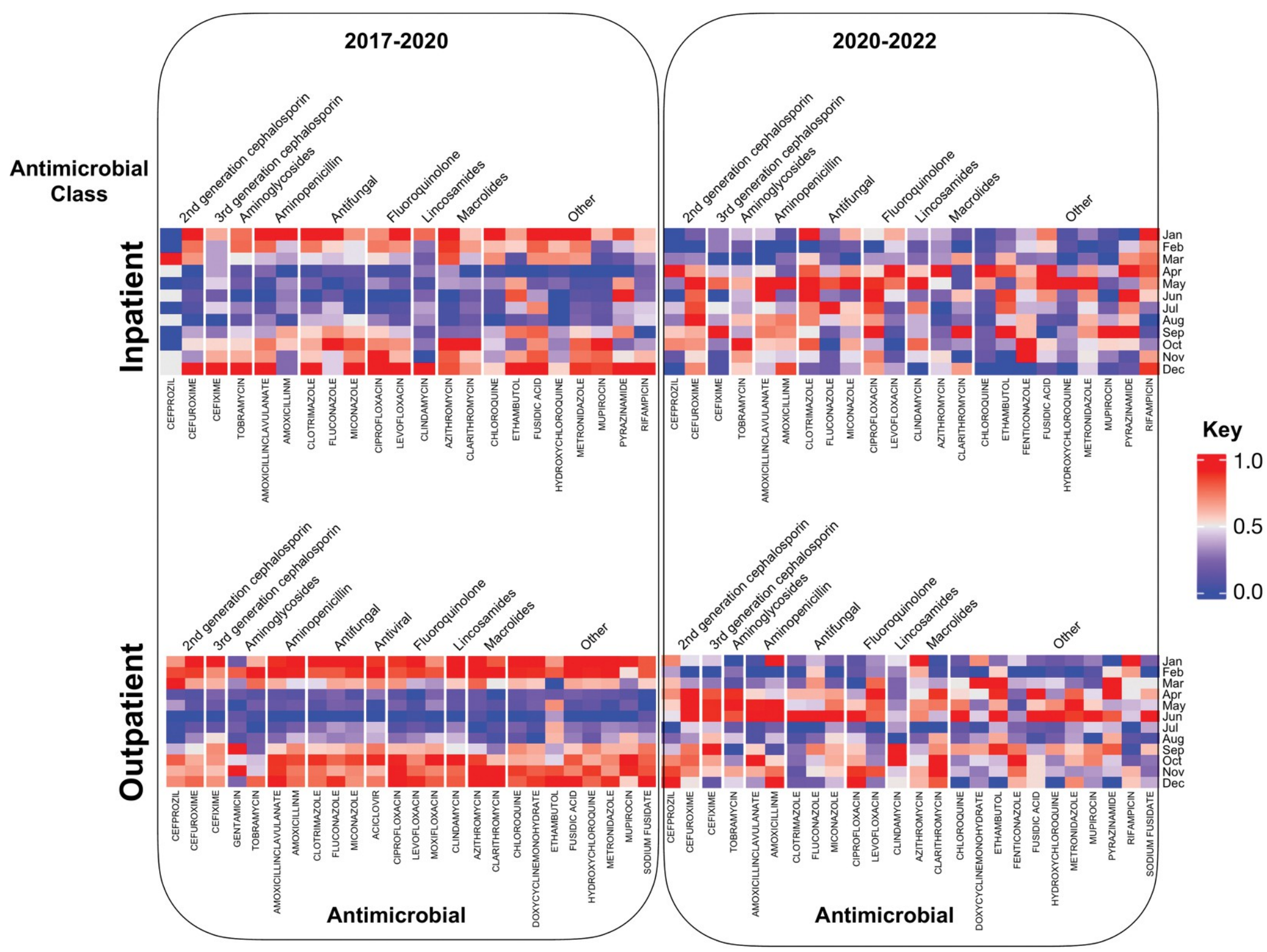


**Fig 1. Heatmap summarizing seasonal distribution of antimicrobial prescriptions for inpatients and outpatients for the pre- and post-covid era for the top 25 highly prescribed antimicrobials in inpatients and outpatients.** Each column has been normalized based on the maximum and minimum values.


antimicrobials out of 20 antimicrobials for inpatients and outpatients shifting from fall-winter to spring-summer, respectively (Fig 1).

## Prescription pattern and co-morbidity

The seasonal alterations in prescriptions for multiple antimicrobials are also apparent in the symptoms diagnosed among both inpatient and outpatient populations. Systematic text analysis of medical records and extraction of relevant medical terms from the comorbidity corpus for each drug reveals that for prescriptions after 2020, terms related to COVID-19 (i.e. "covid" and "coronavirus") were among the top ten most frequently occurring comorbidity terms for 12 out of 20 most commonly prescribed antimicrobials for inpatients, indicating widespread use of multiple antimicrobials with the emergence of COVID-19 (Fig 2A and 2B). However, these terms were less prominent among outpatient groups and were not found among the top ten comorbidity terms, indicating a higher implication of COVID-19 for antimicrobial prescriptions in hospitalized patients. Although less prominent, the importance of COVID-19 related terms for outpatient antimicrobial prescriptions was also found to be consistent to that of inpatients (Fig 2C). For both groups, the sudden shifts in prescription patterns due to COVID-19 appear to encompass a wide range of drugs, including those used for bacterial and fungal infections (e.g. linezolid, cefixime, nystatin, and miconazole) as well as drugs that were used specifically for treatment of COVID-19 infections during the early phase of the pandemic (e.g. azithromycin and chloroquine).

## Temporal clustering of prescription patterns

We identified trends in antimicrobial prescription via clustering analysis (Fig 3). Temporal changes appeared significant for most antimicrobials and fall under four and five clusters with distinctive patterns for inpatients and outpatients, respectively (Fig 3B and 3C). Prescription

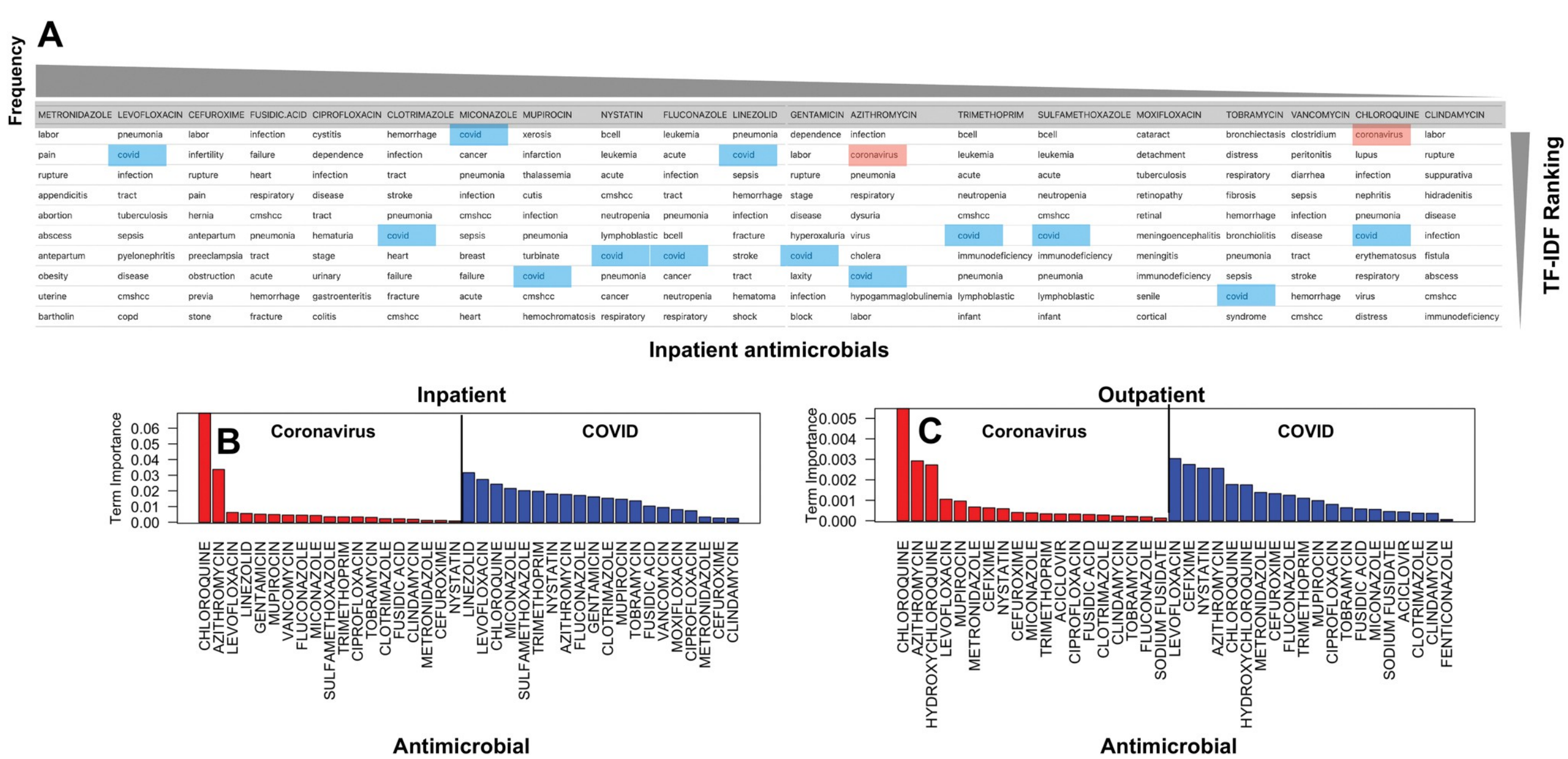


**Fig 2.** A) The top ten most important terms extracted from the admission diagnosis notes for the mostly prescribed antimicrobials for inpatients. The highlighted terms are COVID related terms of <<COVID>>(blue) and <<Coronavirus>>(red). Frequency refers the frequency of the prescription of the antimicrobials. The importance of the top-ranking terms, as measured by tf-idf, for the most frequently prescribed antimicrobials among B) inpatients and C) outpatients.


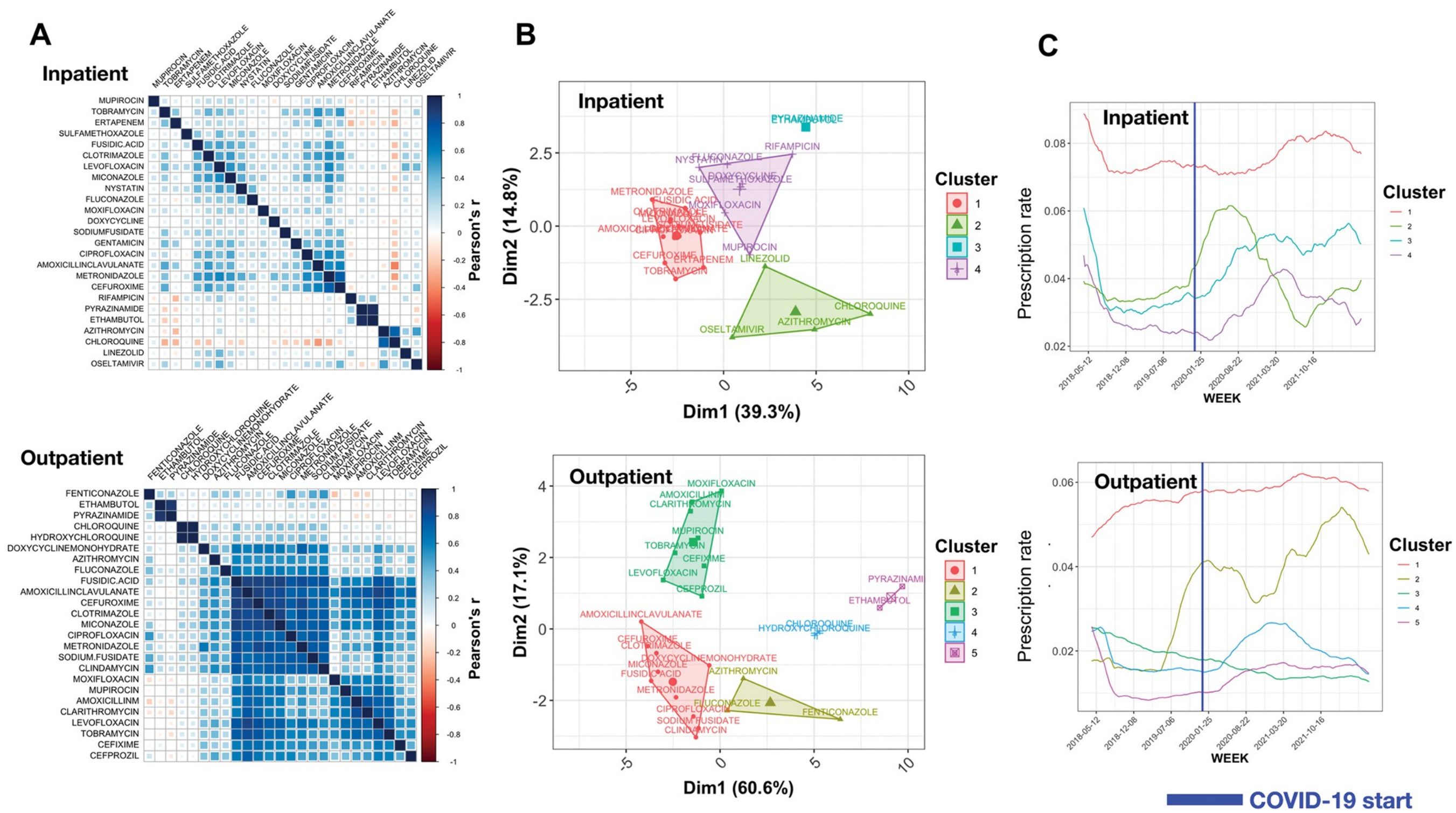


**Fig 3. Major temporal trends in antimicrobial prescriptions.** A) The pairwise correlation between the temporal prescription trends for inpatient and outpatient antimicrobials. The colour intensity corresponds to the strength of correlation. B) The clustering of prescription trends with parametric modelling approach. C) The ensemble trends of antimicrobials for the groups. Each line is an average of the antimicrobial trends in each cluster. COVID-19 start corresponds to the month when the first case of SARS-CoV-2 was reported.


of antimicrobials belonging to cluster 2 for inpatients and clusters 2 and 4 for outpatients showed a rapid increase after the onset of COVID-19, followed by a sudden drop. These clusters contain azithromycin, and chloroquine, drugs which were thought to be effective against COVID-19 but their initial use declined as evidence for their lack of efficacy emerged [33,34]. The intervention times series analysis for azithromycin for inpatient prescriptions indicated a significant evident effect of COVID-19 with a maximum measured effect of 17% increased prescription in the 36-week windows before and after pandemic onset (Fig 4A and 4B). In time window longer than 48 weeks COVID-19 effect was reversed (Fig 4B). For azithromycin prescription in outpatients, the COVID-19 effect gradually increased and was found to level off at 30% in time windows including 120 weeks before and after the pandemic (Fig 4B). For chloroquine, in both inpatients and outpatients prescriptions, the COVID-19 effect was maximized the 18-week windows (inpatient) and the 42-week windows (outpatient), with the effect of 15% (inpatient) and 41% (outpatient), but it dropped after 21 (inpatient) and 84 (outpatient) weeks (Fig 4B). While the prescription rate of antimicrobials in the abovementioned clusters dropped in both inpatients and outpatients around six months after COVID-19, corresponding to the end of the first wave of COVID-19 in June 2020, the rate slightly increased again for cluster 2 for inpatients (Fig 3C). Although the increase is not comparable to the first wave, the findings suggest amendments made in the prescription of these drugs, which were consistent with updates to treatment guidelines. Other antimicrobials in these clusters include linezolid and the antifungals fenticonazole and fluconazole, which are presumably used for COVID-19-associated fungal infections [35]. For linezolid, an effect of 30% was observed in the

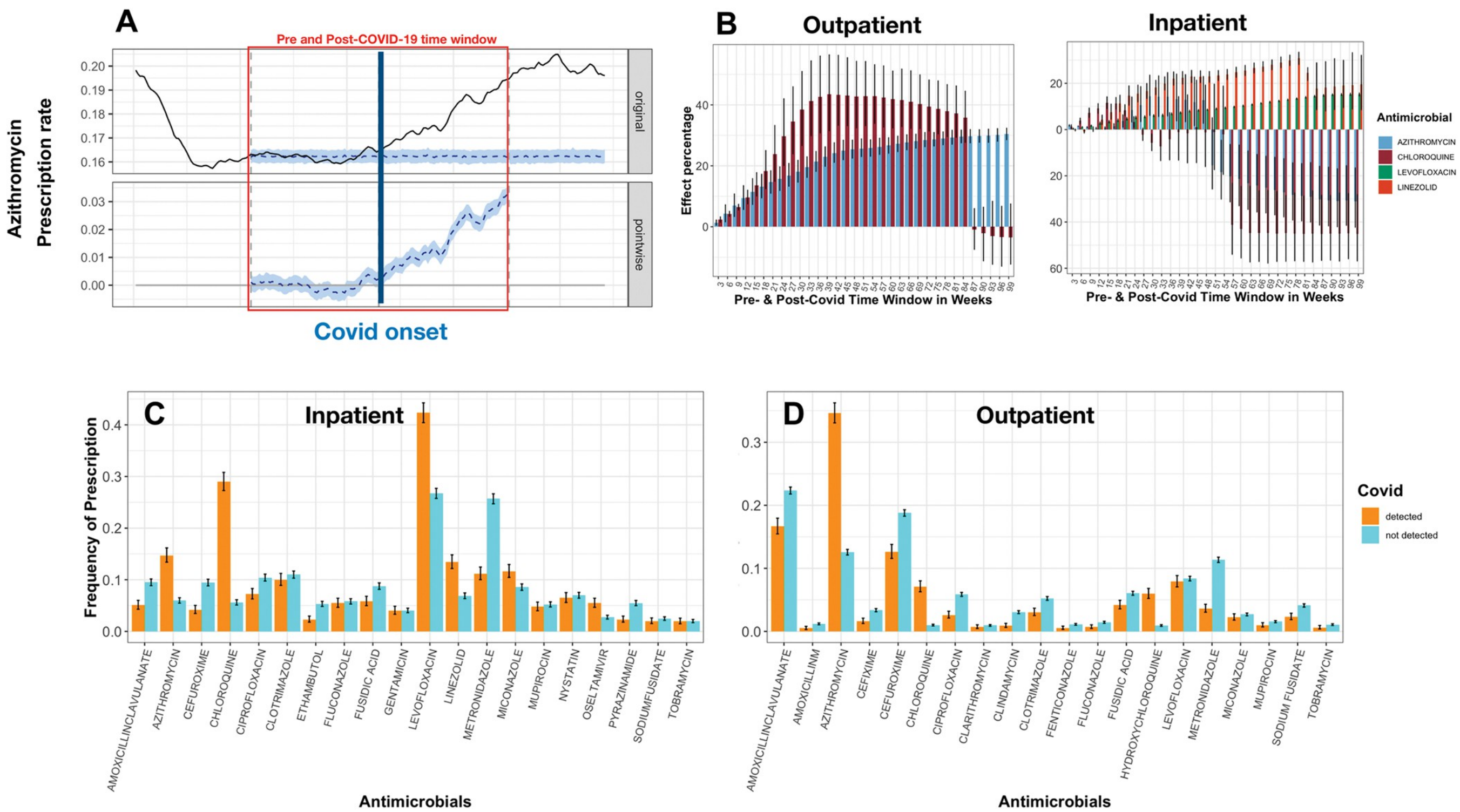


**Fig 4.** The rate of prescription of drugs across the inpatient A) The measured effect of COVID-19 incidence on Covid rate for Azithromycin. The upper panel shows the real values. The dotted line corresponds to mean value of the baseline case in which COVID-19 had not occurred. The dotted line in the bottom panel shows the mean inferred causal effect of COVID-19. The grey area corresponds to 95% confidence interval. B) The inferred causal effects for major antimicrobials that showed significant differences in prescription between patients with and without SARS-CoV-2. C) and outpatient settings D) for patients with and without a confirmed COVID-19 test. The error bars denote 95% confidence intervals.


78-week windows before and after the onset of the pandemic for inpatient prescriptions (Fig 4B). Clusters 3 for inpatients and 5 for outpatients included the anti-TB drug pyrazinamide. The groups show a consistent increase followed by slight drops. As the rate of TB infection did not show a marked increase after COVID-19, as per the reports, the rise might be attributable to an inappropriate prescription (Fig 3B and 3C) [36]. Clusters 1 and 3 for inpatients and clusters 1 and 4 for outpatients included most drugs and showed a declining trend before the onset of COVID-19. Except for cluster 4 in outpatients, for which the rate of decline either decreased or levelled off after COVID-19, for the other three groups a steady increase after COVID-19 was observed. The initial sudden drop has been reported in various settings and is attributable to various disruptions in the supply chain of antimicrobials or antimicrobial treatment plans, but the long-term and lasting prescription increase is evident for most classes of antimicrobials in these clusters. The intervention times series for levofloxacin belonging to cluster 1 for inpatients showed a steady increase over time after COVID-19, equivalent to 10% up to 60 weeks, respectively, after the onset of COVID-19, showing the lasting inclusion of the drug in prescriptions (Fig 4B). The consistent post-COVID-19 prescription increase for the drugs in these clusters is also reflected by the importance of COVID-related terms in the diagnosis notes (Fig 2). For the inpatient prescriptions, out of 18 antimicrobials in these groups, 12 contained COVID-related terms as a top diagnostic term, confirming the over-prescription pattern across various classes of antimicrobials.

## Antimicrobial prescription and COVID-19 infection

The increase in prescriptions could be attributable to secondary bacterial co-infections in patients with COVID-19. Therefore, to assess the extent to which COVID-19 infection is linked with antimicrobial prescriptions, we examined antimicrobial prescription rate patients with and without confirmed COVID-19 over the same time frame for the most prescribed antimicrobials. Despite the consistent trend of increasing prescription rate, the association with COVID-19 rate showed marked differences across antimicrobials (Fig 4C and 4D). As expected, azithromycin, chloroquine and hydroxychloroquine prescription were found to be on average 3.5 times higher in COVID-19 patients than in non-COVID-19 patients. Levofloxacin and the antifungal miconazole were over-prescribed in COVID-19 positive patients but this was only evident among inpatients and not outpatients. Consistent with the prescription notes in Fig 3A, a two-fold increase in prescriptions of linezolid was observed in COVID-19 positive patients. The prescription of 4 and 11 antimicrobials (belonging to a range of classes) was higher in COVID-19 negative patients compared to COVID-19 positive patients in inpatients and outpatients respectively (Fig 4C and 4D). No significant difference was found for 10 antimicrobials in outpatients and 8 antimicrobials in inpatients (Fig 4C and 4D).

## Utilization of antimicrobials and resistant strains

The examination of the association between drug prescription and elevated resistance to the same drugs in major organisms revealed 11 and 10 combinations of drugs and organisms in inpatient and outpatient groups respectively, for which resistance data for the same/respective drug was available (see Methods). The drugs included ciprofloxacin, azithromycin, amoxicillin/clavulanate, cefuroxime, and fusidic acid. The odds ratio for resistance linked to prescription was on average 1.3 (range 0.5–2.7) for inpatients and 1.2 (range 0.5–1.75) for outpatients (S2 Fig, bottom panel). After controlling for demographic factors (age, BMI, gender and ethnicity), the positive association between prescription and resistance remained significant for 7/11 combinations in outpatients and 6/10 in inpatients, with the strongest effects seen for ciprofloxacin in both Gram-negative (*E. coli*) and Gram-positive (MRSA) strains (S2 Fig, top panel). To further understand the connection between resistance level and antimicrobial utilization rate per patient, we also evaluated the cross-correlation between the time series data for the utilization pattern and the resistance level. This was intended to determine the strength of correlation and the time lag at which the effect of prescription could be observed on the resistance level. Fig 5A depicts an example of a positive correlation between the antimicrobial utilization rate in outpatients and the resistance level for erythromycin in *E. coli*, which is significantly detectable with a one-week lag (p-value Pearson correlation $< 0.01$). Our results confirmed a significant positive correlation between the curves at different time lags, specifically for 8 out of 11 combinations of drugs and organisms for inpatients and 9 out of 10 combinations for outpatients, respectively (Fig 5B). This positive slope between the curves indicates that an increase in the consumption rate per patient has a positive effect on the resistance level of the antimicrobials after one week, and this effect is consistently maintained in the subsequent weeks (Fig 5B). However, despite the predominantly positive correlation, the increasing or decreasing pattern was not consistent across all drugs and organisms, such as ciprofloxacin in *K. pneumoniae* and *E. coli* among inpatients (Fig 5B), which is likely due to differences in the course and duration of antimicrobial treatments. Overall, these findings support the connection between antimicrobial prescription and resistance levels at the population level.

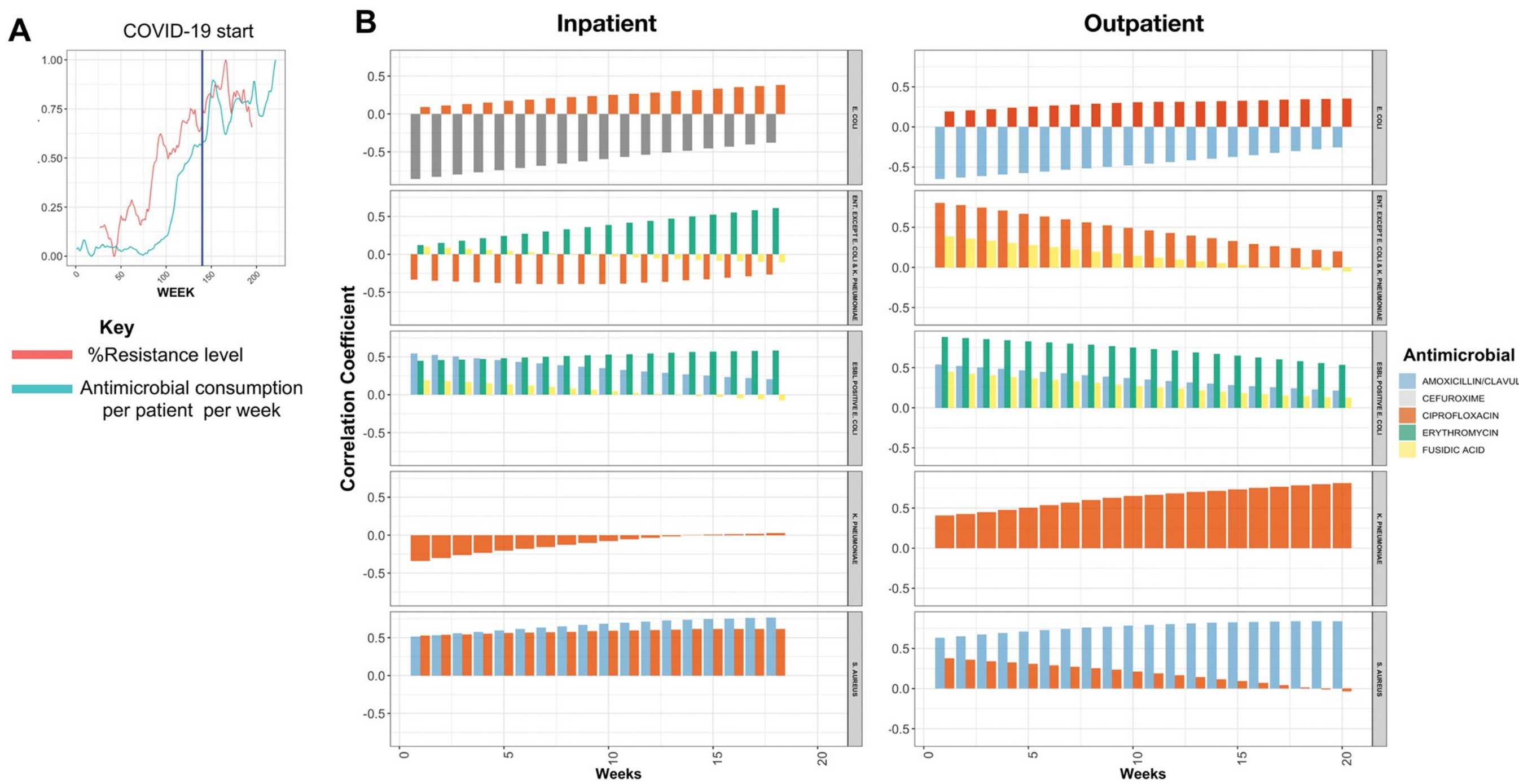


**Fig 5. The correlation between time series for antimicrobial weekly consumption rate and resistance level.** A) The blue and red curves correspond to the consumption rate and resistance level for macrolides in *E. coli* infections. X and Y axis were normalized based on the maximum and minimum values to allow comparison between the time series. B) The correlations between the timeseries as measured by the sample cross-correlation function (CCF). The negative and positive values for the regression refer to the time series lags or leads for the resistance level, respectively.


## Association of antimicrobial resistance with mortality

We examined the lethality of resistance strains at the population level. In a drug susceptibility test of 100 combinations of the ten most prevalent organisms and the ten most commonly prescribed antimicrobials for each organisms, 82 conditions (i.e., drug and organism pairings) were found to have colonization of patients with both resistant and susceptible strains. Of these conditions, 51/82 (62%) of conditions turned out to show a significant link between mortality and antimicrobial resistance ($p$-value $< 0.05$), in which an odds ratio greater than one was observed in 49/51 (96%) of the conditions (Fig 6A). This significant link persisted after adjusting for the confounding factors (see Methods) for 38/49 (77%) of conditions (i.e., drug and organism pairings) from the logistic regression analysis and for 24/49 (%48) conditions from the competing risk survival analysis (estimated coefficient significantly greater than zero in Fig 6A). For Gram-negative pathogens, a higher number of resistant antimicrobials was associated with higher mortality. This was particularly evident for *K. pneumoniae* and *E. coli drug-resistant* strains, where resistance against 85% of drugs were found to be positively linked with higher mortality (odds-ratio values greater significantly than one in Fig 6A), and to a lesser extent MRSA and *P. aeruginosa* (Fig 6A). Moreover, a high mortality odds ratio was observed with resistance to second and third generation cephalosporins, such as cefuroxime and ceftazidime, with an average increase of 15%, which is comparable to the estimates from randomized clinical trials [37]. The highest odds ratio was found for carbapenem-resistant pathogens, with an odds ratio of 7.5 and 4.7 for *K. pneumoniae* and *E. coli*, respectively (Fig 6A). The presence of ESBL and carbapenemase-producing genes, i.e. $bla_{\text{CTX-M}}$, $bla_{\text{NDM}}$ and $bla_{\text{OXA-48}}$, showed a positive association with mortality, which remained significant for strains harboring $bla_{\text{NDM}}$ and $bla_{\text{OXA-48}}$ in the logistic regression analysis and for $bla_{\text{OXA-48}}$ in

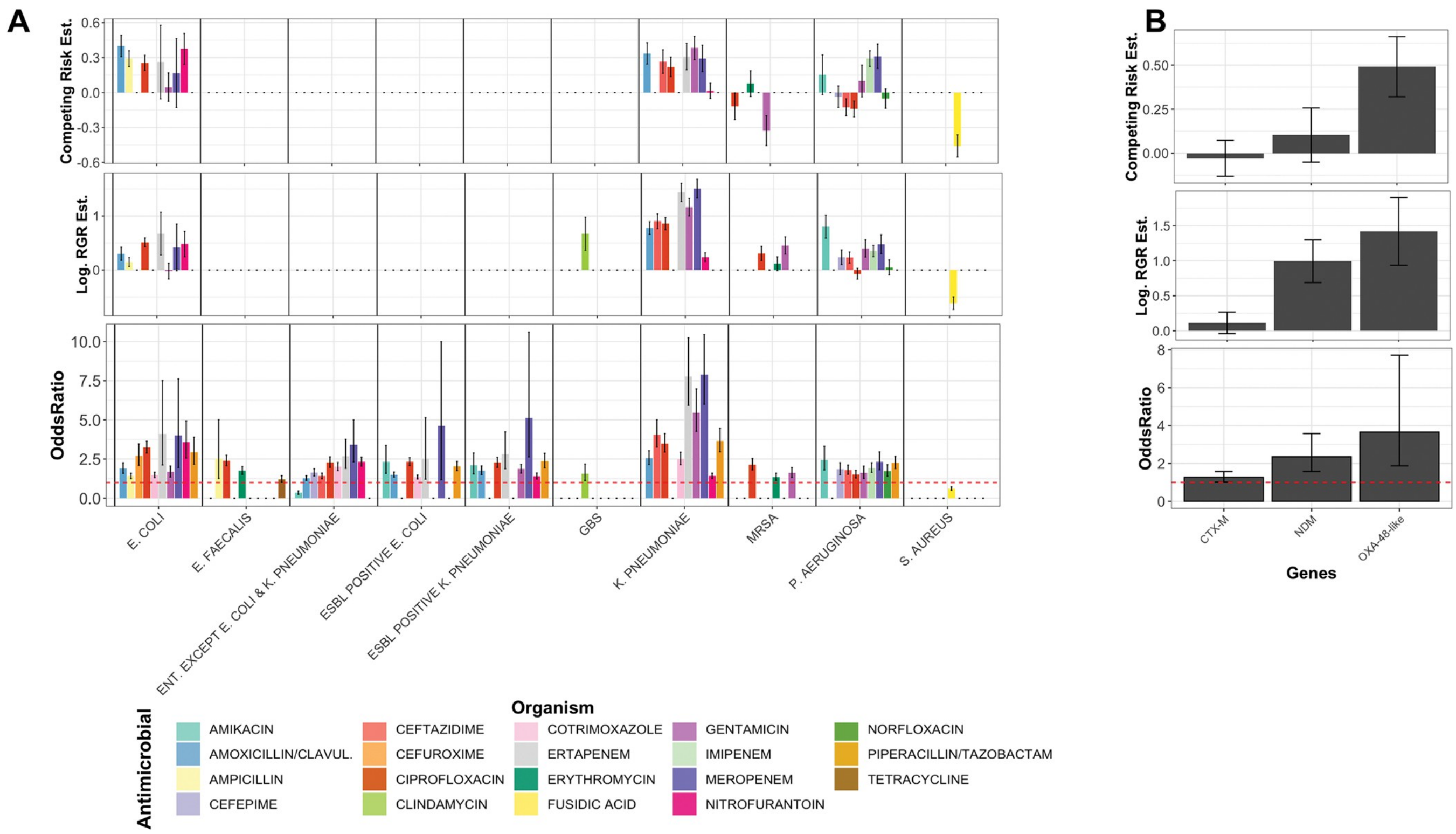


**Fig 6.** The odds of death across A) various groups of antimicrobials and organisms and B) the presence of major carbapenemase and ESBL genes. Only significant values are shown. The upper panels in A) and B) show the strength of coefficients from the survival analysis and logistic regression analysis after accounting for the impact of the confounders (see Methods). The dotted line in the bottom figure in A) and B) corresponds to odds ratio of one (no effect).


competing risk survival analysis after controlling for confounding factors (Fig 6B). The mortality rate was higher in the presence of carbapenemase-producing genes (*bla*$_{\text{NDM}}$ and *bla*$_{\text{OXA-48}}$) compared to ESBL genes (*bla*$_{\text{CTX-M}}$). Despite the significant link of colonization by resistant strains, the link between resistance and mortality was not significantly different for COVID-19-positive and COVID-19-negative patients (data provided in GitHub directory for the project).

## Antimicrobial resistance pattern

The integration of prescription and antibiogram data revealed a short-term impact of prescription rates on the resistance level. To thoroughly analyze the changes in resistance, we initially selected the antimicrobials and organisms exhibiting fluctuations in resistance over a five-year period (see Methods). Out of 194 combinations of drugs and organisms for which complete resistance data was available, 80/194 and 128/194 pairs recorded changes exceeding 10% and 5% respectively in resistance levels over the five-year period. An upward trend was observed for several groups of second and third generation cephalosporins, carbapenems, and macrolides in both Gram-positive and Gram-negative strains (S3 Fig). The resistance patterns fell under six generic trends (Fig 7). Four out of six clusters showed an upward trend after COVID-19, reflecting the effect of the prescriptions of antimicrobials (Fig 7A and 7B). Each group was composed of a range of drugs and covered Gram-positive and Gram-negative organisms, possibly due to the co-selection of resistance across different groups (Fig 7C). Group 1, consisting of second and third generation cephalosporins and gentamicin for

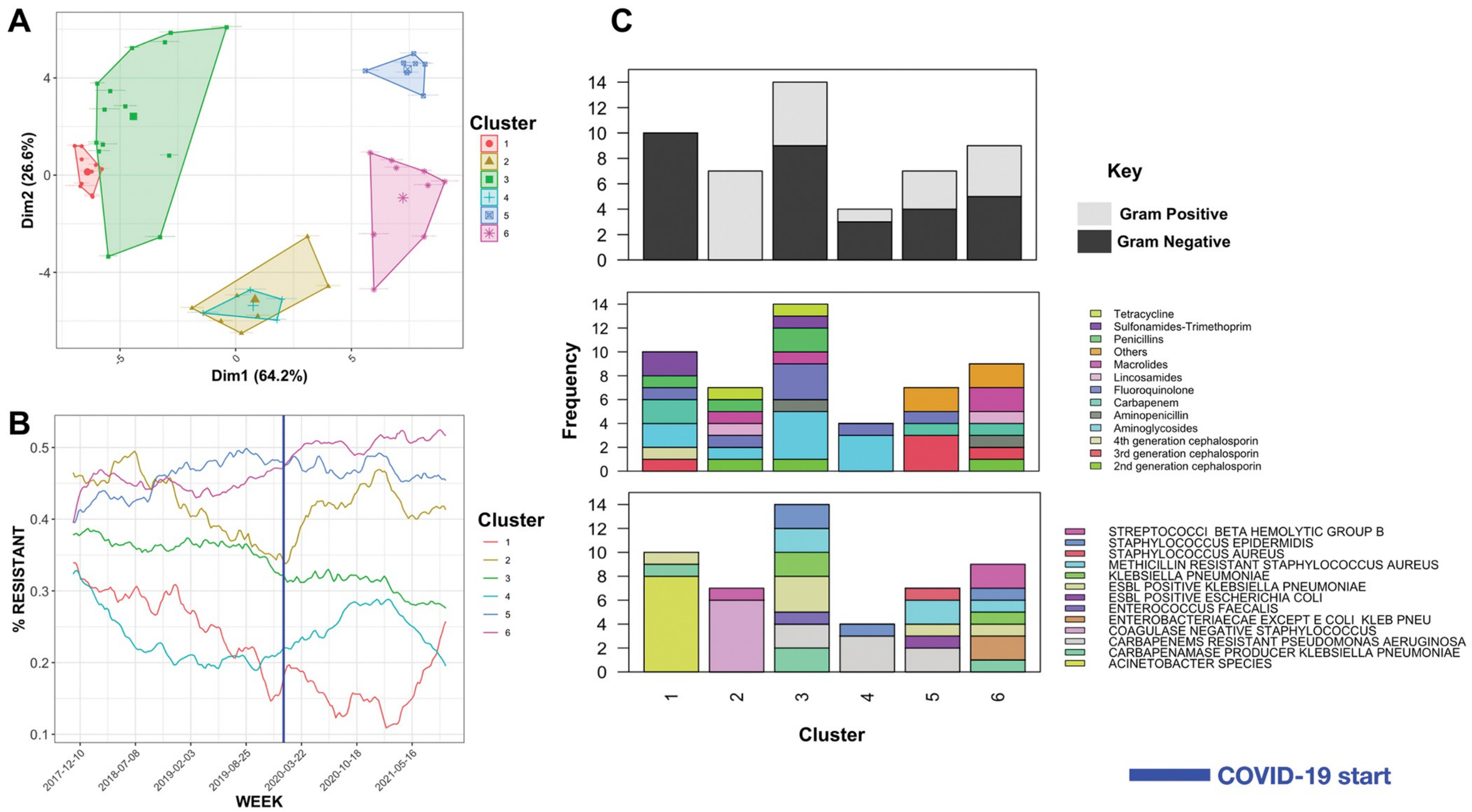


**Fig 7.** A) Major clusters in antimicrobial resistance across strains and for different drugs. B) Major trends over time the blue vertical line shows the start of COVID-19 C) the distribution of organisms and drugs in the clusters.


Acinetobacter, showed a decline that became more pronounced after the COVID-19 pandemic but gradually increased after 2020 (Fig 7B). Group 3 displayed a stable pattern with a steady decrease after the COVID-19 pandemic, which could indicate a longer-term effect of the pandemic. Group 5, which includes fusidic acid and ceftazidime, saw a sharp increase during 2017 but remained stable thereafter. Groups 2 and 4 revealed a rapid increase after the onset of the COVID-19 pandemic, particularly evident in Group 2, composed of staphylococcal strains, which may be a result of the use of antimicrobials for treating secondary infections in COVID-19 patients. Group 4, composed of Gram-negative carbapenem resistant strains including *P. aeruginosa* showed a sudden peak in resistance after the onset of the COVID-19 pandemic, which followed a prior decrease in resistance levels. These findings indicate the dynamic and co-selection patterns of resistant strains that hold across different antimicrobials.

## Concordance of resistance trends among organisms

A comparative analysis of resistance to commonly used antimicrobials has revealed a high degree of concordance among different organisms. For the four common drugs for which complete records were available for multiple organisms, the average correlation between the temporal trends of resistance was 0.73. The resistance level of the macrolide erythromycin was similar across three Gram-positive staphylococcal and streptococcal pathogens, with a rapid peak observed after the COVID-19 pandemic, which is likely due to an increase in the use of the related antimicrobial azithromycin (S4 Fig). The third-generation cephalosporin ceftazidime showed a monotonically increasing pattern in four out of five strains (S4 Fig). In contrast, greater divergence in resistance trends was observed for the aminoglycoside gentamicin

and the fluoroquinolone ciprofloxacin, even for the species belonging to the same class. Notably, the resistance pattern for *Klebsiella* differed from the other Enterobacteriaceae strains, potentially due to complex population structures and dominant clones in clinical settings [38]. Nevertheless, the overall concurrence of trends suggests a consistent population-level resistance across strains, indicating the concurrent evolution of resistance against different classes of antimicrobials.

## Discussion

Using a comprehensive set of large-scale clinical datasets, we analyzed the patterns of antimicrobial utilization and resistance in a large hospital network and assessed the effects of COVID-19 on these patterns. Our analysis of the comprehensive 5-year dataset (2017–2022) of clinical, diagnostic laboratory and prescription records revealed a significant and sudden shift in the use of antimicrobial drugs for both inpatient and outpatient populations, with some antimicrobials experiencing prolonged effects up to two years after the start of the COVID-19 pandemic. We also found a high mortality rate associated with infections caused by Gram-negative carbapenem resistant pathogens and a rapid impact of antimicrobial consumption on resistance patterns, observable within a few weeks. These findings offer comprehensive insights into the interplay between the evolution of AMR and the COVID-19 pandemic.

The prescription pattern showed an overlap in the antimicrobials most frequently prescribed in the outpatient and inpatient settings. In addition, the observed pattern and high prescription frequency of beta-lactam and fluoroquinolone antibiotics is similar to those reported in other settings [32], These similarities are likely due to the consistency between treatment guidelines across countries and are indicative of the commonality of challenges in the implementation of antibiotic stewardship during the pandemic. A significant shift in the timing of peak occurrence of prescriptions from fall-winter pre-pandemic to spring-summer during the pandemic was observed for outpatients in the USA [22]. The pre-pandemic peak timing is consistent with the known seasonal increment of antimicrobials during the winter months and the influenza season. The spring-summer peak during the pandemic was consistent across multiple groups of antimicrobials suggesting that multiple factors, such as the overlap of bacterial pneumonia symptoms with those of COVID-19 [23], empirical antibiotics use for co-infections in COVID-19 patients [24] and the reduced incidence of other respiratory viral infections due to the pandemic non-pharmaceutical interventions, may be drivers for such shift.

The observed significant impact of COVID-19 on antimicrobial prescription practices, is demonstrable in both short-term and long-lasting over-prescription of these drugs. These findings are in keeping with previous reports which highlighted the potential short- and long-term effects of the pandemic on antimicrobial utilization and emergence of resistance [39–42]. Factors such as the heightened use of empiric antibiotics for COVID-19 patients, higher prescription rates of antibiotics for telemedicine consultations and adverse effect on the implementation of antibiotic stewardship programs have been identified [39,40].

Our findings identified significant trends in antimicrobial resistance, which exhibit fluctuations for various drugs and organisms, with an overall increasing trend in resistance levels, especially post-COVID-19. These findings are indicative of the connection between antimicrobial utilization and resistance levels at the population level and demonstrate the impact of prescription patterns per patient on driving the early emergence of resistance which is sustained over time. Although resistance to antimicrobials increased the odds of mortality to an average of 2.5, the link between resistance and mortality was not significantly different for COVID-19

positive and COVID-19 negative patients. Thus, infections caused by antimicrobial resistant strains remain significant cause of concern with likelihood for poor clinical outcome irrespective of the COVID-19 infection.

The findings from this study highlight the utility of large-scale clinical datasets and EHR in studying antimicrobial resistance and provide useful population-level insights into the epidemiology of resistance that could inform policy and practice recommendations. Our work represents an EHR-based method and shows the value of the inclusion of such a dataset for inferring population-level trends in infectious diseases. The use of such datasets remains rare [15], although they have been showcased primarily with data from hospitals in the UK [43,44]. We showed how such databases can provide an understanding of trends in antibiotic prescribing and bacterial infections at a population level. We particularly showed the strength of natural language processing for understanding unstructured prescription text data, in addition to structured data. This not only showed the relative implication of COVID-19 infection for the prescription of antimicrobials but also complemented the ICD-10 diagnostic codes, which is a common limitation in EHR data [15]. Furthermore, the access to standardized microbiology culture and antibiotic susceptibility results across multiple sites, which are among the most difficult EHR elements for institutions, provided us with a high-quality dataset to directly approximate the mortality impact of resistance [45,46]. The impact of the COVID-19 pandemic on the landscape of AMR remains a significant ongoing area of research interest. The patterns of antimicrobial usage and resistance in this study align well with the wide-spread empiric prescriptions of antimicrobials during the pandemic. The novel findings of the impact of the COVID-19 pandemic in driving and sustaining AMR evolution at the population level represent important addition to the literature and may be generalizable to other settings.

Our study provides valuable insights into the epidemiology of AMR, however, it has several limitations. Firstly, the sample only included patients with confirmed bacterial infections and excluded those without infections and neglected antimicrobials prescriptions in patients without any confirmed infections. Additionally, the study was confined to the public hospital network in Dubai and did not include private hospitals in the city (see Methods). The data collected from electronic health records (EHR) may also be biased, as only clinically significant symptoms are recorded [43]. Furthermore, it is well documented that a significant portion of antimicrobial prescriptions lack proper justification or connection to a clinical condition, and the use of antimicrobials in the community may not be adequately controlled [47,48]. These limitations may be addressed by including more comprehensive datasets from the private sector, as recommended before (Lord O'Neill report's 2016 report). While our study focused solely on healthcare facilities in Dubai, the expansive and comprehensive nature of our dataset suggests the potential generalizability of our findings, as demonstrated earlier for consumption and resistance patterns. This suggests that the insights gained might have some applicability in other contexts. However, the availability of EHR data on antimicrobial consumption and resistance is still expanding. Consequently, to further discern shared characteristics and unique elements, future comparative studies with other locations are required.

In conclusion, antimicrobial resistance was a global complex health issue before the onset of the pandemic. The onset of the COVID-19 pandemic added a new dimension to the AMR landscape, by modifying the processes of emergence, transmission, and infection burden. Although the pandemic is now under control, the long-term effect of the pandemic on AMR is yet to be determined. Dissecting this effect requires efforts to integrate antimicrobial stewardship into pandemic responses. The novel approach of integration of the epidemiological and clinical datasets in this study allowed us to identify some of the key factors driving the evolution of AMR at the population level. Future studies could leverage predictive modelling using information within datasets to forecast future trends. Such systems could be readily integrated

into EHR systems to not only improve decision-making for prescriptions but also to support interventional and surveillance studies to contain the dissemination of resistance.

## Supporting information

**S1 Fig.** The prescription patterns A) The prescription of drugs in- and out-patients with the names of antimicrobials exclusive to either group of patients. B) The frequency of top 25 highly prescribed antimicrobials for in- and out-patient groups.
(DOCX)

**S2 Fig.** The link between prescription rate and resistance level and the deaths odds ratio for the A) inpatient and B) outpatient groups. The frequency of prescriptions for the resistant and susceptible strains for the prescription of the same drugs across the organisms. The upper panel shows the odds-ratio for resistance within the patients with the prescription for the antimicrobials.
(DOCX)

**S3 Fig. The distribution of resistance levels for drugs and organisms for which more than 10% difference over time was observed in resistance level. The overall trend for antimicrobials obtained from start and end points in 2022 and 2017.**
(DOCX)

**S4 Fig. Resistance trends for the same drugs across different organisms.** The blue line shows the onset of COVID-19.
(DOCX)

## Acknowledgments

We acknowledge the support of the Dubai Academic Health Corporation (DAHC) and Dubai Health Authority (DHA). The authors also acknowledge the kind support of the Information Technology Team (formerly of DHA and now DAHC); Dr Farida Alkhaja, Senior Advisor, DAHC, Dr Hussain Al Samt Head of Laboratories, DAHC; Dr Zulfa AlDeesi, Head of Clinical Microbiology Laboratory, Latifa Hospital, DAHC; Dr Mohamed Sameh Ali, Head of Pharmacy, Rashid Hospital, DAHC; and Dr Lina Bahjat, Senior Pharmacist, Rashid Hospital, DAHC.

## Author Contributions

**Conceptualization:** Danesh Moradigaravand, Abiola Senok.

**Data curation:** Danesh Moradigaravand.

**Formal analysis:** Danesh Moradigaravand, Abiola Senok.

**Funding acquisition:** Danesh Moradigaravand.

**Investigation:** Danesh Moradigaravand, Hamda Hassan Khansaheb, Maya Habous, Hanan Alsuwaidi, Alawi Alsheikh-Ali.

**Methodology:** Danesh Moradigaravand.

**Project administration:** Danesh Moradigaravand, Alawi Alsheikh-Ali.

**Resources:** Abiola Senok, Laila Al-Dabal, Hamda Hassan Khansaheb, Maya Habous, Hanan Alsuwaidi, Alawi Alsheikh-Ali.

**Supervision:** Danesh Moradigaravand, Alawi Alsheikh-Ali.

**Validation:** Danesh Moradigaravand, Abiola Senok.

**Visualization:** Danesh Moradigaravand.

**Writing – original draft:** Danesh Moradigaravand, Abiola Senok.

**Writing – review & editing:** Danesh Moradigaravand, Abiola Senok, Alawi Alsheikh-Ali.

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
