## [Decision Letter · Decision Letter 0]

2 Aug 2023

PDIG-D-23-00246

Unveiling the dynamics of antimicrobial utilization and resistance in a large hospital network over five years: Insights from health record data analysis

PLOS Digital Health

Dear Dr. Moradigaravand,

Thank you for submitting your manuscript to PLOS Digital Health. After careful consideration, we feel that it has merit but does not fully meet PLOS Digital Health's publication criteria as it currently stands. Therefore, we invite you to submit a revised version of the manuscript that addresses the points raised during the review process.

Please submit your revised manuscript within 30 days Sep 01 2023 11:59PM. If you will need more time than this to complete your revisions, please reply to this message or contact the journal office at digitalhealth@plos.org. Please include the following items when submitting your revised manuscript:

We look forward to receiving your revised manuscript.

Kind regards,

Mengyu Wang, Ph.D.

Academic Editor

PLOS Digital Health

Journal Requirements:

a. State what role the funders took in the study. If the funders had no role in your study, please state: “The funders had no role in study design, data collection and analysis, decision to publish, or preparation of the manuscript.”

b. If any authors received a salary from any of your funders, please state which authors and which funders.

2. Please provide separate figure files in .tif or .eps format.

3. We noticed that you used “data not shown” in the manuscript. We do not allow these references, as the PLOS data access policy requires that all data be either published with the manuscript or made available in a publicly accessible database. Please amend the supplementary material to include the referenced data or remove the references.

Additional Editor Comments (if provided):

See comments from Reviewer 1. Please make your code available before we can accept your paper. Please also at least release part of your raw data for reproducibility. Only sharing post-analysis data is not helpful.

Reviewers' comments:

Reviewer's Responses to Questions

**Comments to the Author**

1. Does this manuscript meet PLOS Digital Health’s publication criteria? Is the manuscript technically sound, and do the data support the conclusions? The manuscript must describe methodologically and ethically rigorous research with conclusions that are appropriately drawn based on the data presented.

Reviewer #1: Yes

2. Has the statistical analysis been performed appropriately and rigorously?

Reviewer #1: I don't know

3. Have the authors made all data underlying the findings in their manuscript fully available (please refer to the Data Availability Statement at the start of the manuscript PDF file)?

Reviewer #1: No

4. Is the manuscript presented in an intelligible fashion and written in standard English?

Reviewer #1: Yes

5. Review Comments to the Author

Reviewer #1: Unveiling the dynamics of antimicrobial utilization and resistance in a large hospital network over five years: Insights from health record data analysis 

The aim of this work was to look at impact of COVID-19 on antimicrobial resistance (AMR) evolution as well as ecological trends. It was great to see the open source GitHub but this currently has no code for analysis only some summary csv files. It is fine that the individual patient level data is not available I think as it is healthcare data. 

I cannot assess the machine learning analysis, but it seems a reasonable and exciting approach to the free text in the infection diagnosis. I’d also like the authors to check and compare with other methods for looking at the link between resistance and mortality odds: there are some complex statistical methods for looking at competing risks and comorbidities. 

Abstract

- Are the 60551 patients those with infection? Or all registered in the network? If former, then should go after the next sentence

- Results: mortality from infection or all mortality? 

- Results: what prescription practices? New ones? 

- Results: COVI D-19 gap to be removed in last line

- Conclusions: How do they emphasise the long term effect of COVID-19? Just two years / technically still in the pandemic? 

Introduction

Line 33: I’d add “in part” as it may also be driven a lot by transmission of resistant pathogens

Line 49: Expand EHR first time use

Line 55: word missing after “and”

Line 68: missing “of” 

Line 64: what is the coverage of this dataset? For those not aware of the health system in Dubai how many people go to the public vs private healthcare? 

Methods

Line 80: how did you define bacterial infection? This can be difficult (is it infection or just colonisation?) did you use a set code? Was it just bloodstream infections? Was it different for different patient groups / bacteria? 

Line 83: what types of tests are done? MIC / EUCAST etc? has it changed over time? 

Line 89: which strains were sent for molecular typing? 

Line 95: which patients were tested for COVID-19? Only those with symptoms? 

Line 104: did you only get antibiotic prescribing data for those with an infection or all patients? Was this actual prescribing data to individual patients? 

Line 114: linked to above, was it all prescribing or just prescribing in those with infection? 

Line 118: why are you looking at seasonal trends? There should be something in the introduction on the importance of seasonality to antibiotic usage as well as the previous work on lag times between use and resistance 

Line 127: should it be SARS-CoV-2 infection? COVID-19 is the disease. Similarly SARS-CoV-2 positive not COVID-19 positive? 

Line 131: tense error 

Line 150: I can’t asses this machine learning but it’s a nice idea to look at the free text in this structure way

Line 171: did you look at whether the number of tests varied over time? It may be that this could drive resistance prevalence changes 

Line 184: notation not totally clear here 

Line 193: something need in the introduction about what / why look differently at inpatient and outpatient 

Line 195: was the top most frequent tested consistent over time? 

Line 208: shouldn’t it have been done by the type of infection? i.e. those with higher mortality rates should be separate from those with lower rates? 

Line 210: did you account for any of the confounding due to longer hospital stays? 

Line 219: I think this equation really needs to account for the type of infection

Line 220: there is no coefficient beta in the equation

Line 230: there are no analysis codes in here only output data csvs – please add the code or modify this statement

Results

Line 234: what is a “confirmed” infection as opposed to the bacterial infections you talk to in the methods? 

Line 234: is there any uncertainty around these values? It must have changed over time? 

Line 235: what is a “prescription”? is it always the same amount? E.g. 3 days of gentamicin is the same as 7 days? 

Line 238: did you expect this sharing? Something to mention in the intro / discussion

Line 239: how do you define dominant? 

Line 241: checking here: this is all outpatients or just those with infections? 

Line 385: Did you check all associations for confounding factors like length of hospital stay? See here for some discussion https://www.ncbi.nlm.nih.gov/pmc/articles/PMC8210247/

Line 451: does the data support the MDR strains? Did you look at antibiograms? I.e. strain profiles not just individual antibiotic resistance prevalences? 

Discussion

Line 471: can you add some references here to previous work on seasonal shifts? 

Line 495: I am concerned that these mortality odds have not taken into account all the variability in sampling / length of stay and other comorbidities. 

Line 506: I agree that it is relatively uncommon but has been performed by others e.g. 

Especially in the UK e.g. 

https://www.ncbi.nlm.nih.gov/pmc/articles/PMC9303046/

https://www.microbiologyresearch.org/content/journal/jmm/10.1099/jmm.0.001724

https://pubmed.ncbi.nlm.nih.gov/28333200/

line 528: how many patients go to private hospitals? 

Linked to line 418: how does these trends linking impact of COVID-19 on AMR link to the time lags you saw in the use-resistance time series analysis? 

- How general do you think the results are to Dubai? Can you make any broader conclusions? 

Figure 2: Can you explain why you highlight blue / red in the top? COVID linked to the bottom graph right? 

- What does tf-df stand for? 

- Only inpatient use is shown in the top correct? 

Figure 3:

- Blue line “covid start”: can you be more specific? Was this lockdowns in Dubai? And it is “COVID-19”

Figure 4: 

- Covid should be capitalised

- Is “covid onset” the same as covid start in Figure 3? Please be consistent. And different to “COVID-19 start” in Figure 5

- (c) and (d) legend explanation is missing something? 

Figure 5: 

- A: why does resistance rise before rising consumption levels? 

- Line 640: “for” the resistance level? 

Figure 6:

- Why show both the estimate and the odds ratio? How do they relate? Are they both accounting for ethnicity age bmi and gender? 

Figure 7: 

- Is A trends or clusters? B is trends?

6. PLOS authors have the option to publish the peer review history of their article (what does this mean?). If published, this will include your full peer review and any attached files.

**Do you want your identity to be public for this peer review?** For information about this choice, including consent withdrawal, please see our Privacy Policy.

Reviewer #1: No

---

## [Decision Letter · Decision Letter 1]

1 Dec 2023

Unveiling the dynamics of antimicrobial utilization and resistance in a large hospital network over five years: Insights from health record data analysis

PDIG-D-23-00246R1

Dear Dr Moradigaravand,

We are pleased to inform you that your manuscript 'Unveiling the dynamics of antimicrobial utilization and resistance in a large hospital network over five years: Insights from health record data analysis' has been provisionally accepted for publication in PLOS Digital Health.

Best regards,

Mengyu Wang, Ph.D.

Academic Editor

PLOS Digital Health

Reviewer Comments (if any, and for reference):

Reviewer's Responses to Questions

**Comments to the Author**

1. If the authors have adequately addressed your comments raised in a previous round of review and you feel that this manuscript is now acceptable for publication, you may indicate that here to bypass the “Comments to the Author” section, enter your conflict of interest statement in the “Confidential to Editor” section, and submit your "Accept" recommendation.

Reviewer #1: All comments have been addressed

2. Does this manuscript meet PLOS Digital Health’s publication criteria? Is the manuscript technically sound, and do the data support the conclusions? The manuscript must describe methodologically and ethically rigorous research with conclusions that are appropriately drawn based on the data presented.

Reviewer #1: (No Response)

3. Has the statistical analysis been performed appropriately and rigorously?

Reviewer #1: (No Response)

4. Have the authors made all data underlying the findings in their manuscript fully available (please refer to the Data Availability Statement at the start of the manuscript PDF file)?

Reviewer #1: (No Response)

5. Is the manuscript presented in an intelligible fashion and written in standard English?

Reviewer #1: (No Response)

6. Review Comments to the Author

Reviewer #1: (No Response)

7. PLOS authors have the option to publish the peer review history of their article (what does this mean?). If published, this will include your full peer review and any attached files.

**Do you want your identity to be public for this peer review?** For information about this choice, including consent withdrawal, please see our Privacy Policy.

Reviewer #1: None
